



# Integrating wide swath altimetry data into Level-4 multi-mission maps

Maxime Ballarotta[1], Clément Ubelmann[2], Valentin Bellemin-Laponnaz[3], Florian Le Guillou[4], Guillaume Meda[1], Cécile Anadon[1], Alice Laloue[1], Antoine Delepoulle[1], Yannice Faugere[5], Marie-Isabelle Pujol[1], Ronan Fablet[6], Gérald Dibarboure[5]

[1]Collecte Localisation Satellites (CLS), Ramonville-Saint-Agne, France
[2]Datlas, Saint Martin d'Hères, France
[3]Université Grenoble Alpes (UGA), Grenoble, France
[4]European Space Agency (ESA), Frascati, Italy
[5]Centre National d'Études Spatiales (CNES), Toulouse, France
[6]IMT Atlantique, Plouzané, France

*Correspondence to*: M. Ballarotta (mballarotta@groupcls.com)

**Abstract.** Real-time observation of ocean surface topography is essential for various oceanographic applications. Historically, these observations relied mainly on satellite nadir altimetry data, which were limited to observe scales greater than approximately 60 km. However, the recent launch of the wide-swath SWOT mission in December 2022 marks a significant advancement, enabling the two-dimensional global observation of finer oceanic scales (~15 km). While the direct analysis of the two-dimensional content of these swaths can provide valuable insights into ocean surface dynamics, integrating such data into mapping systems presents several challenges. This study focuses on integrating the SWOT mission into multi-mission mapping systems. Specifically, it examines the contribution of the SWOT mission to both the current nadir altimetry constellation (six/seven nadirs) and a reduced nadir altimetry constellation (three nadirs). Our study indicates that within the current nadir altimetry constellation, SWOT's impact is moderate, as existing nadir altimeters effectively constrain surface dynamics. However, in a hypothetical scenario where a reduced nadir altimetry constellation is envisioned to be operational by 2030, the significance of wide-swath data in mapping becomes more pronounced. Alternatively, we found that data-driven and dynamical mapping systems can significantly participate in refining the resolution of the multi-mission gridded products. Consequently, integrating high-resolution ocean surface topography observations with advanced mapping techniques can enhance the resolution of satellite-derived products, providing promising solutions for studying and monitoring sea-level variability at finer scales.

## 1 Introduction

Real-time observation of ocean surface topography is crucial for various oceanographic applications (marine navigation, marine safety, climate research...). Since the 1990s, in addition to in-situ observations, the use of nadir radar altimetry has revolutionized operational oceanography and enhanced our understanding of ocean surface dynamics on a global scale (Le Traon, 2013). Over the years, advancements in altimetry technologies and increased sampling have significantly improved the



precision and accuracy of altimetry products, and the wide-swath altimeter/interferometer technology currently being tested
during the SWOT mission has the potential to significantly enhance oceanographic observation.

The SWOT mission (Fu and Rodriguez, 2004: Morrow et al., 2019), resulting from a collaboration between NASA, the Centre
National d'Études Spatiales (CNES), the Canadian Space Agency (CSA), and the UK Space Agency, was launched in
December 2022. The main objective of the SWOT mission is to observe and track water surface elevation on Earth for the first
time in 2D and with unprecedented resolution. Unlike conventional altimetry missions, SWOT relies on a wide-swath, the
KaRin instrument, enabling the observation of fine oceanic surface topography scales with a resolution up to 15 km, whereas
conventional altimeters are often limited to resolutions of 60 km (Dufau et al., 2016, Vergara et al., 2023). In addition to the
swath, SWOT also features a nadir altimeter on board.

Nadir (1D) and KaRin (2D) altimetry data can sometimes pose complexities for the oceanographic community, which prefers
the use of gridded, spatially, and temporally continuous data, such as Level-4 multi-mission maps (e.g., AVISO/DUACS multi-
mission maps (Ducet et al, 2000), MEaSUREs maps (Beckley et al., 2010), GLORYS12v1 maps (Lellouche et al., 2021)).
These maps are produced through interpolation, modelling, and/or assimilation techniques, and have been regarded as the
reference for monitoring and understanding the ocean surface dynamics over the last three decades. However, resolving
variability at length scales smaller than approximately 150-200 km with these mapping methods is challenging (Ballarotta et
al., 2019). Operational oceanography now seeks to incorporate increasingly finer spatial and temporal scales into its products,
including, for example, mesoscale and sub-mesoscale dynamics.

Advancing our understanding of fine-scale ocean processes is essential for improving ocean models, predictions, and gaining
deeper insights into ocean surface dynamics. Although large meso-scale eddies have been extensively studied over the past 30
years using multi-mission gridded altimetry products and are identified as key contributors to the horizontal transport of heat,
nutrients, and carbon (Dong et al., 2014, Zhang et al., 2014), submesoscale variability and its role in ocean dynamics remain
poorly observed and understood. The SWOT mission thus represents an excellent opportunity to better understand the role of
submesoscale eddies, enhance the spatial resolution of the altimeter products and paves the way for new challenges in the
utilization, validation, and integration of these data into mapping systems. Since April 2023, the SWOT mission can be
integrated into the current altimetry constellation including Jason-3, Sentinel-3A, Sentinel-3B, SARAL/Altika, Cryosat-2 and
Haiyang-2B (Fig. 1). Until now, only simulations through Observing System Simulation Experiments (OSSE) have been
conducted to study the contribution of wide-swath systems (like SWOT) to mapping systems (e.g., Benkiran et al., 2021,
Tchonang et al., 2021, Ubelmann et al., 2015, Le Guillou et al., 2021). In this study, we aim to investigate and assess the
impact of these new wide-swath data on a global mapping system through the use of Observing System Experiments (OSE).
The main objective of this study is to quantify the contribution of wide swath in mapping systems.



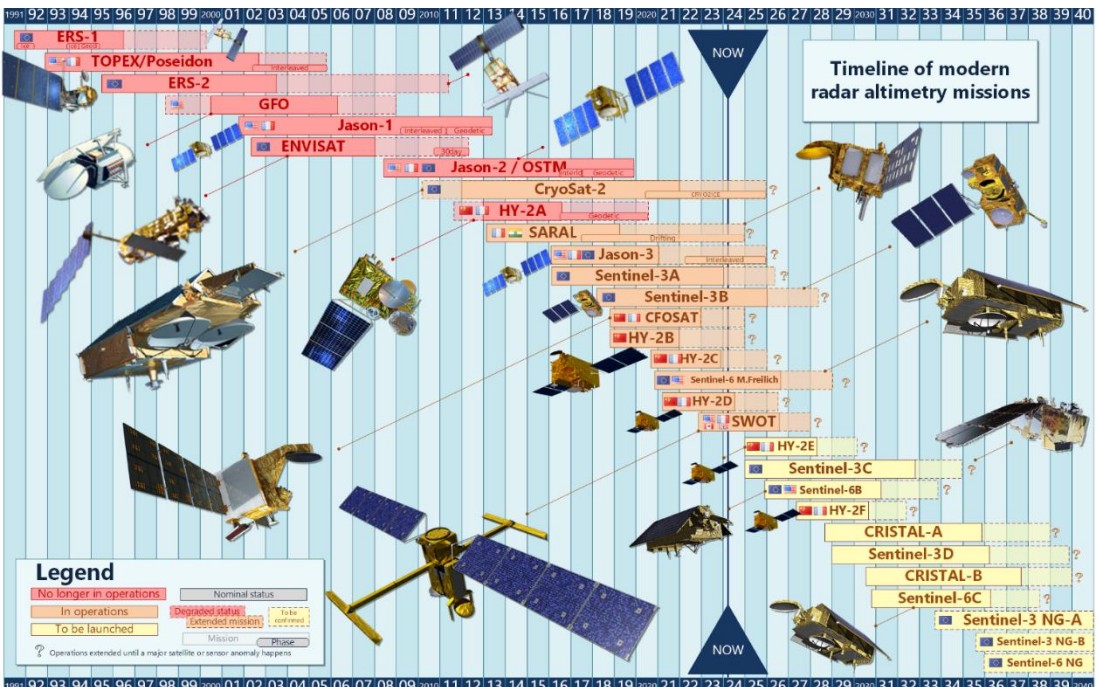

**Figure 1: Past, present, and future altimetry constellation. From Aviso+ (2022). Timeline of modern radar altimetry missions. https://doi.org/10.24400/527896/A02-2022.001**

The paper is organized as follows: Section 2 outlines the data sources and mapping techniques used in this study. Section 3 presents the experiments and validation metrics. The key results are then presented in section 4. Lastly, the potential of wide swath altimetry in both contemporary and future altimeter constellations is discussed, along with the current limitations observed in our mapping process. A discussion on the benefits and limitations of OSSE is given in an Appendix as well as a

75 first analysis of alterative regional mapping methods.

## 2 Data and methods

### 2.1 Data

The mapping method used in this study takes input data from several remote sensing observations, which are summarized in Table 1 and described below.



**Table 1: Data used in the study.**

| Product type | Nadirs Sea-level anomaly Level 3 products | SWOT Sea-level anomaly Level 3 products |
|---|---|---|
| Product ref. | SEALEVEL_GLO_PHY_L3_NRT_008_044 | SWOT_L3_SSH |
| Spatial coverage | [0°E:360°E] [90°S:90°N] | [0°E:360°E] [90°S:90°N] |
| Temporal coverage | From 2023-07-01 to 2024-05-15 | From 2023-07-27 to 2024-05-01 |
| DOI | https://doi.org/10.48670/moi-00147 | https://doi.org/10.24400/527896/A01-2023.018 |

### 2.1.1 Nadirs sea-level anomaly Level 3 products

To produce the gridded sea level maps, we used the global ocean sea level anomaly observations from the Near-Real-Time (NRT) Level-3 altimeter satellite along-track data distributed by the EU Copernicus Marine Service (product reference SEALEVEL_GLO_PHY_L3_NRT_008_044, Pujol et al., 2023), specifically for the Jason-3, Sentinel-3A, Sentinel-3B, Sentinel-6A, SARAL-Altika, Cryosat-2, Haiyang-2B, missions. This dataset covers the global ocean and is available at a sampling rate of 1 Hz (approximately 7 km spatial spacing). The quality of this dataset is ensured through the implementation of homogenization and cross-validation procedures aimed at eliminating residual orbit errors, long-wavelength errors, large-scale biases, and discrepancies among different data streams. A description of the geophysical and environmental corrections applied to the dataset can be found in the quality information document (Pujol et al., 2023) and is summarized in Eq. (1). In this study, we focus on unfiltered sea-level anomalies (SLAs) corrected with dynamic atmospheric correction (dac), and ocean tide.

$$\text{SLA} = \text{Orbit} - \text{Range} - \sum(\text{Environmental Corrections}) - \sum(\text{Geophysical Corrections}) \\ - \text{Mean Sea Surface} \tag{1}$$

with $\sum$ (Environmental Corrections) = wet tropospheric + dry tropospheric + ionospheric + sea-state-bias, $\sum$ (Geophysical Corrections) = solid earth tide + load tide + ocean tide + pole tide + dynamic atmospheric correction. The spatial coverage of the nadir altimeter constellation is illustrated in Figure 2a for a reduced constellation of 3 nadirs and Figure 2b for a constellation of 6 nadir altimeters. Specifically, the figure shows differences in sampling density within the various constellations.



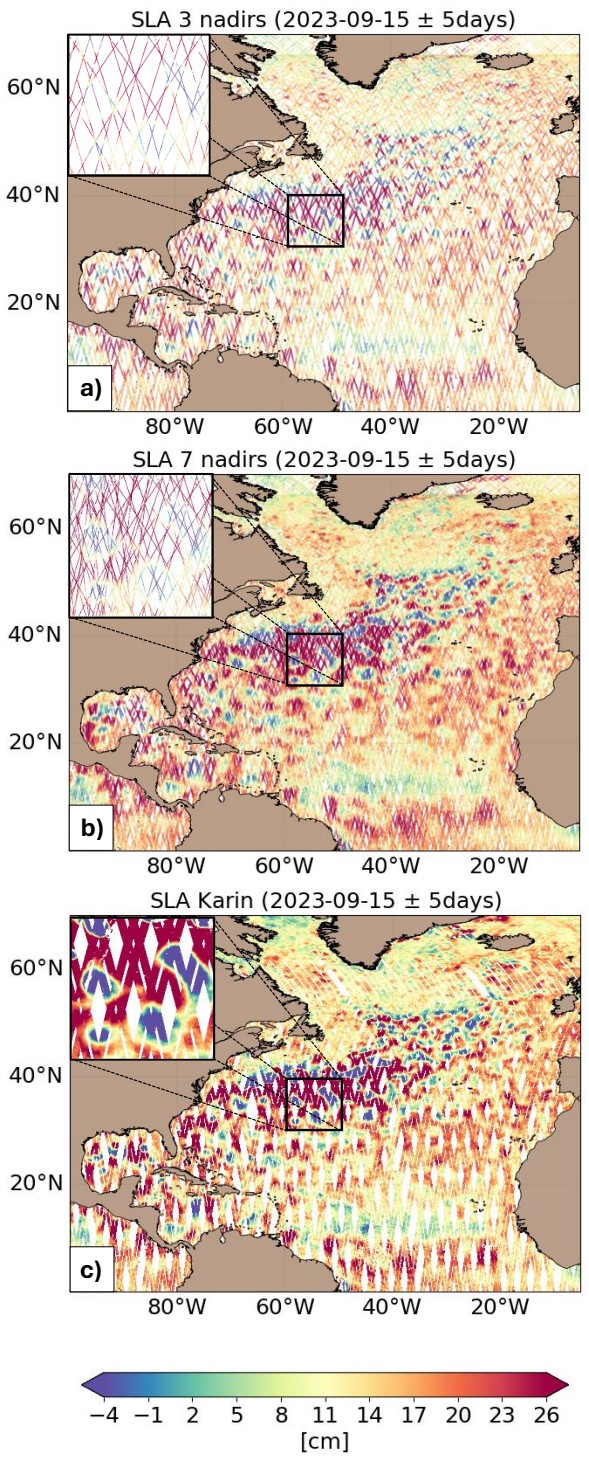

**Figure 2: Spatial sampling of a) 3 nadirs altimeters, b) 7 nadirs altimeters and c) 1 SWOT.**



### 2.1.2 SWOT sea-level anomaly Level 3 products

In addition to the nadir altimetry constellation previously mentioned, we conducted experiments involving the integration of SWOT Level-3 Ocean product (specifically referencing SWOT_L3_SSH) during the 21-day phase of the mission. The SWOT_L3_SSH product combines ocean topography measurements collected from both the SWOT KaRIn and nadir altimeter instruments, consolidating them into a unified variable on a 2 km spatial grid spacing. For our investigation, we used version 1.0 of the product accessible through the AVISO+ portal (AVISO/DUACS, 2023). These data were derived from the Level-2 "Beta Pre-validated" KaRIn Low Rate (Ocean) product (NASA/JPL and CNES). The processing methodology for SWOT Level 3 products is described in Dibarboure et al. (2024), following a sequence including Level-2 correction (e.g., homogenization with other satellite in term of geophysical correction…), editing (e.g., detection of spurious measurements), and multi-mission calibration (data-driven calibration to mitigate residuals biases between all sensors in the constellation). The spatial coverage of the SWOT instrument is shown in Figure 2c, demonstrating high spatial sampling in certain regions (particularly at high latitudes), as well as moderate spatial sampling in other regions due to low revisit times at those latitudes.

### 2.2 Mapping method

The global maps produced in this study are based on the Multiscale Inversion of Ocean Surface Topography (MIOST) technique (Ubelmann et al. 2021, 2022). This mapping approach is specifically designed to manage large volumes of observational data, such as those from the 2km SWOT Level-3 Karin data at the global scale, and to capture various modes of variability in ocean surface topography and currents. MIOST achieves this by constructing multiple independent components within an assumed covariance model. In this study, the focus is on the geostrophic mode to represent the geostrophically balanced evolution of sea surface height (SSH).

The quality of the MIOST gridded products has been accessed through both idealized and real observational systems (Ubelmann et al., 2021, 2022, Ballarotta et al., 2023), demonstrating their effectiveness to map global surface topography and currents. This study employs Delayed-Time (DT) mode processing, which integrates both past and future observations to enhance the interpolation process.

Similar to the optimal interpolation techniques used in operational context (e.g., Le Traon et al, 1998, 2003; Ducet et al., 2000; Pujol et al., 2016), MIOST operates within a linear and gaussian framework. Additionally, experimental regional mappings were conducted using non-linear approaches, such as the 4DvarQG (Le Guillou et al., 2024) and 4dvarNET approach (Fablet et al., 2021) approaches, with a detailed evaluation of these techniques provided in Appendix B.



## 3 Experiments and validation metrics

### 3.1 Experiments

We conducted several experiments (as summarized in Table 2) to investigate the impact of wide swath altimetry in the mapping
constrained by present-day and future altimeter constellations.

**Table 2: List of experiment carried in this study.**

| | Input data | |
|---|---|---|
| **Experiment** | **Nadir altimeters** | **SWOT** |
| EXP1: 6 nadirs | All w/o Altika | No |
| EXP2: 6 nadirs + SWOT | All w/o Altika | **Yes** |
| EXP3: 3 nadirs | Sentinel3A, Sentinel3B, Sentinel6A | No |
| EXP4: 3 nadirs + SWOT | Sentinel3A, Sentinel3B, Sentinel6A | **Yes** |

In the baseline experiment (EXP1), SSH maps covering the period from July 27, 2023, to May 1st, 2024, were generated using
data from six altimeters: Jason-3, Cryosat-2, Sentinel-3A, Sentinel-3B, Sentinel-6A, and Haiyang-2B. To ensure independent
evaluations, one altimeter (Saral/AltiKa) was omitted from the mapping process. This experiment is representative of the
present-day nadir-only altimeter constellation. A second experiment (EXP2) aimed to evaluate the integration of SWOT into
the existing altimeter constellation. SSH maps for the same timeframe were generated using data from the six aforementioned
altimeters along with SWOT L3 SSH data. Similar to EXP1, Saral/AltiKa data was excluded for independent assessments. A
third experiment (EXP3) was conducted considering a reduced altimeter constellation scenario comprising only three
altimeters (Sentinel-3A, Sentinel-3B, Sentinel-6A). Finally, a fourth experiment (EXP4) was conducted to assess the
integration of SWOT in a future altimeter constellation. For this experiment, data from three altimeters (Sentinel-3A, Sentinel-
3B, Sentinel-6A) and SWOT L3 SSH data were considered.

### 3.2 Validation metrics

The validation metrics are based on statistical and spectral analysis.

One quantitative assessment involves comparing SSH maps with independent SSH along-track data. This diagnostic follows
3 main steps: 1) Interpolating the gridded SSH data to the locations of the independent along-track SSH; 2) Calculating the
mapping error $SSH_{error} = SSH_{map} - SSH_{alongtrack}$ and 3) Performing a statistical analysis on the $SSH_{error}$. Prior to the statistical
analysis, a filter may be applied to focus on specific spatial scales, such as the 65km to 200km range, which is relevant for
short mesoscale signal and representative of the scale on which SWOT is expected to have a significant impact. The validation





metrics is based on the error variance scores in 1°x1° longitude x latitude boxes (or averaged over specific region of interest), defined as:

$$\sigma_{err} = \frac{\sum_{t=1}^{N}\left(SSH_{error}(x,y,t) - \overline{SSH_{error}(x,y,t)}\right)^2}{N} \tag{2}$$

Where $x$ is the longitudinal position of an along track measurement, $y$ the latitudinal position of an along track measurement, $t$ the time position of an along track measurement, $N$ is the total number of SSH measurement in the box (or area of interest) and the overbar indicate the sample statistical mean.

The comparison of the error variance score between two experiments informs about the gain or reduction Δ of the mapping error, for example:

$$\Delta = \sigma_{err}(EXP2) - \sigma_{err}(EXP1) \tag{3}$$

The previous diagnostic was conducted in physical space. For a wavelength-specific assessment and to avoid spatio-temporal filtering issues, diagnostics can be performed in frequency space using spectral analysis. As described for example in Ballarotta et al. (2019), this involves:

1) Interpolating the gridded SSH data to the locations of the independent along-track SSH
   2) Dividing the data into segments (1500km long segment)
   3) Storing these segments with their median coordinates
   4) Performing spectral analysis of the segments found in 10°x10° boxes

For each box, we compute the mean power spectral densities of the independent signal (SSH$_{alongtrack}$) and the mapping error (SSH$_{map}$- SSH$_{alongtrack}$). The signals are detrended and windowed with a Hanning function before spectral calculation. The signal-to-noise ratio (SNR, Equation 4) is derived from the power spectral densities, and the effective resolution is determined as the wavelength λs where the SNR(λs) is 2 (Equation 5), i.e., the wavelength where the SSH$_{error}$ is two times lower than the signal SSH$_{alongtrack}$.

$$SNR(\lambda) = \frac{PSD(SSH_{along-track})(\lambda))}{PSD(SSH_{error})(\lambda)} \tag{4}$$

$$SNR(\lambda s) = 2 \tag{5}$$

## 4 Results

### 4.1 Qualitative assessment

The impact of integrating L3 SWOT Karin data into the MIOST system is qualitatively illustrated in Figure 3 with a snapshot
of the magnitude of geostrophic current calculated from the SLA map for August 31, 2023. The current intensity appears to be underestimated in the mapping products (Fig 3c & 3d) compared to the intensity derived from the SWOT Karin products



(Fig 3b). Due to its filtering properties, the mapping method cannot resolve the fine-scale filaments present in the Karin data. Nevertheless, the difference between maps constructed with 1 SWOT + 6 nadirs and those constructed with only 6 nadirs (Fig 3e) reveals that some structures are more intense in the maps using the Karin data. Additionally, the positions of these structures are more precisely constrained by the 2D Karin data, leading to better-defined fronts and structures.

**Figure 3: Example of geostrophic current reconstruction on 2023-08-31 with MIOST at a) global scale, b) view from Karin L3 products over the Agulhas region, c) from MIOST reconstruction integration 1SWOT and 6 nadirs, d) from MIOST reconstruction integration 6 nadirs and e) the difference in MIOST reconstructions between integration 1SWOT and 6 nadirs vs 6 nadirs only**

## 4.2 Contribution of wide swath altimetry in the present-day altimeter constellation

The mapping errors arising from using the present-day altimeter constellation (EXP1) are shown in Figure 4. Within EXP1, the most significant error in SSH mapping peaks at 50 cm² (7cm rms) in the western boundary surface current and along continental plateaus (Fig. 4a and b). In offshore regions with low SSH variability, the error variance remains below 10 cm²



(3cm rms). Figures 4c and d illustrate the difference in mapping error between EXP2 and EXP1 for all spatial scales and those smaller than 200km, respectively. A blue (red) pattern indicates a decrease (increase) in mapping error when incorporating SWOT into the mapping process. For all spatial scales, SWOT helps to reduce mapping errors, notably at mid-latitudes, with an average decrease ranging from 5% to 10%. The most significant decrease (∼10%) takes place in regions characterized by
high SSH variability. Mapping results with and without SWOT exhibit similar outcomes in coastal, offshore low variability, and equatorial regions.

For spatial scales smaller than 200km, the relative impact of SWOT is more prominent as expected from the short spatial scales observed by SWOT. This leads to a reduction in mapping errors by approximately 14%, particularly pronounced in regions with high SSH variability. Table 3 outlines the comparative outcomes across various regions of interest (coastal, equatorial
band, low-variability region, and high-variability region). Overall, the integration of the SWOT data mainly contributes to reduce mapping errors in energetic ocean currents such as the Gulfstream, the Kuroshio, and the Antarctic Circumpolar current. Certain regions are prone to degradation in mapping quality when integrating SWOT, particularly those characterized by tropical rainfall and wet troposphere, as well as areas affected by storm tracks or internal tides.

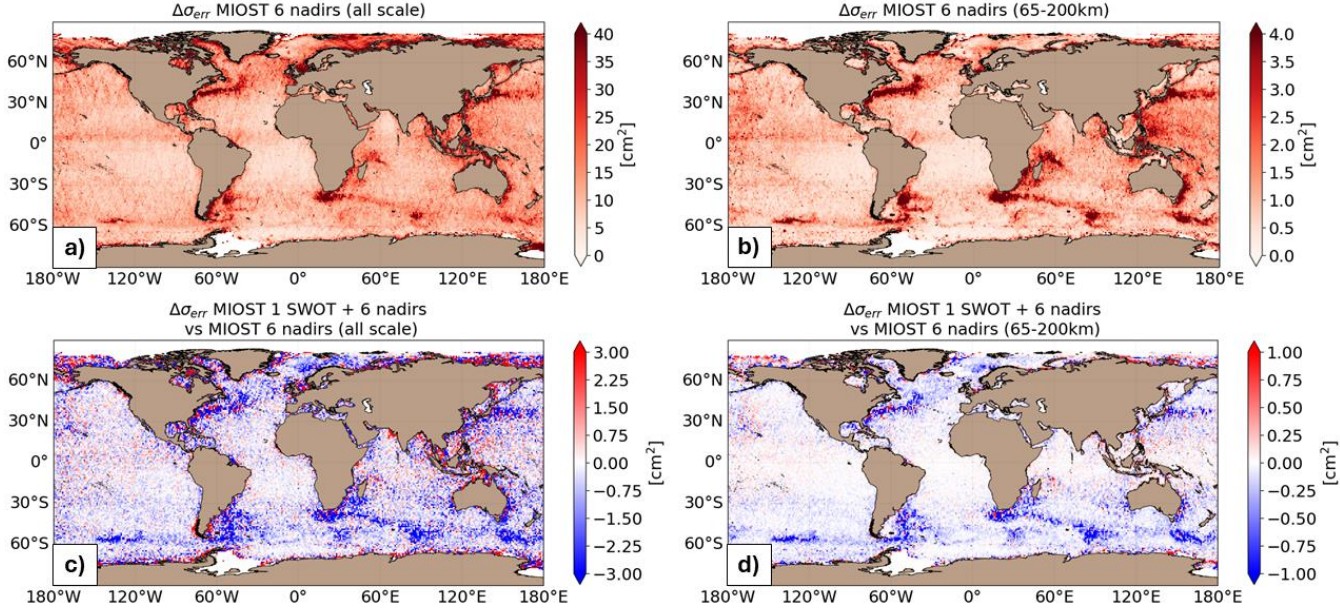


**Figure 4: Variance of the difference SSH$_{map}$-SSH$_{alongtrack}$ computed for the EXP1 (mapping 6 nadirs) and in considering a) all spatial scale and b) spatial scale between 65 km and 200 km. Gain/loss of the mapping error variance of SLA in EXP2 relatively to the EXP1 mapping error variance for c) all spatial scale and d) scale between 65 km and 200 km. Blue colour means a reduction of error variance when SWOT is included in the mapping.**




**Table 3: Regionally averaged mapping error variance for EXP1 and EXP2. Score in brackets is the gain/reduction of error variance on the SSH variable between EXP2 and EXP1.**

|  | **EXP1: 6 nadirs** | | **EXP2: 1SWOT + 6 nadirs** | |
| --- | --- | --- | --- | --- |
|  | Err all scale [cm$^2$] | Err 65-200km [cm$^2$] | Err all scale [cm$^2$] | Err 65-200km [cm$^2$] |
| Coastal (< 200km) | 28.3 | 2.2 | 28.9 (+2%) | 2.2 (0%) |
| Offshore (> 200km) high variability (> 200cm$^2$) | 24.2 | 3.5 | 21.7 (-10%) | 3.0 (-14%) |
| Offshore (> 200km) low variability (< 200cm$^2$) | 11.2 | 1.0 | 10.8 (-4%) | 0.9 (-10%) |
| Equatorial band (10°S-10°N) | 10.6 | 1.1 | 10.6 (0%) | 1.1 (0%) |

Figure 5a illustrates the effective resolution of maps generated using six nadir altimeters, while Figure 5b shows the enhancement in resolution achieved through the integration of SWOT into the mapping process. Maps produced with six nadir altimeters have resolutions ranging approximately from 100 km at high latitude to 500 km in the equatorial region. SWOT contributes to refine map resolutions at mid-latitudes, resulting in local improvements of up to 20 km and an average enhancement of around 5-10 km (see Figure 5b). Degraded resolutions with SWOT are found in the equatorial band and North

Pacific basin.

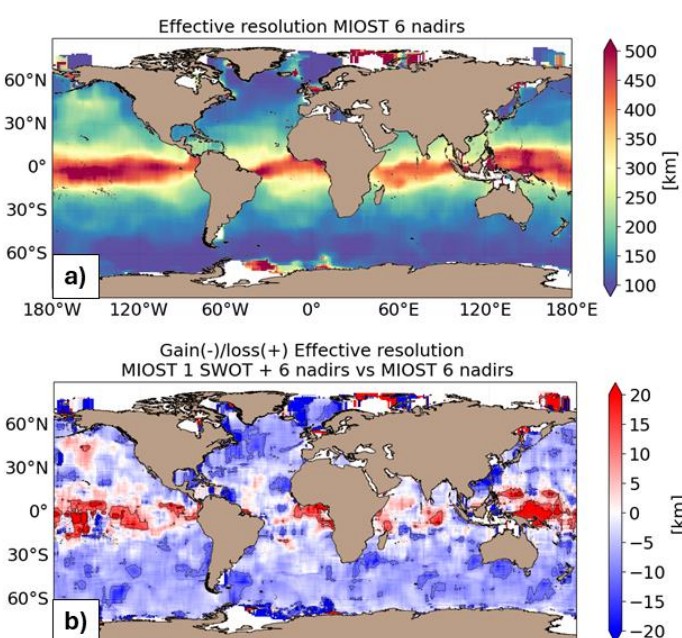

**Figure 5: Maps of effective spatial resolution (in km) for (a) the EXP1 and (b) gain/loss of effective resolution (in km) between EXP2 and EXP1. Blue means finer resolution when SWOT is included in the mapping.**






## 4.3 Contribution of wide swath altimetry in a future altimeter constellation

The mapping errors in using a constellation of 3 nadir altimeters (EXP3) are shown in Figure 6. In this reduced nadir altimeters constellation, the mapping errors are more than 25% larger in high variability region and about 10% larger in other regions compared to a 6 nadirs altimeter constellation mapping. The largest SSH mapping error are found in the western boundary surface current and over the continental plateaus (Fig. 6a and b). In the offshore low-variability region, the error variance is between 10 cm$^2$ to 20 cm$^2$ (3 to 5 cm rms). Figure 6c and d show the difference in mapping error between the EXP4 and EXP3 for all spatial scales and the spatial scales smaller than 200km, respectively. A blue (red) pattern means a reduction (increase) of the mapping error when SWOT is included in the mapping. As for the 6-nadir altimeter constellation, for all spatial scales considered, SWOT allows to reduce the mapping errors, especially at mid-latitudes. The largest reduction in mapping error (∼ 26 %) is found in regions of high variability. In the coastal, offshore low variability and equatorial regions, the mapping error reduction is moderate (~10 %) when including SWOT in the mapping. For spatial scales smaller than 200km, SWOT allows to reduce the mapping errors by ∼ 30 % particularly in the high variability region. Table 4 summarizes the results of the comparison over different regions of interest (coastal, equatorial band, low-variability region, and high-variability region). Overall, the SWOT instrument contributes mainly to reduce the mapping error in energetic ocean currents, such as the Gulfstream, the Kuroshio and in the Antarctic Circumpolar current. As for EXP2, certain regions are prone to degradation in mapping quality when integrating SWOT, particularly regions characterized by tropical rainfall and wet troposphere, as well as areas affected by storm tracks or internal tides. Comparing EXP3 (with 3 nadirs) to EXP1 (with 6 nadirs) enables quantification of the advantage of using a 3-nadir altimeter in the mapping process. As shown in Table 4, the mapping errors for maps constructed with 6 nadirs are relatively similar to those for maps constructed with 1SWOT+3 nadirs. This indicates that incorporating a SWOT mission into Level 4 products could provide observational capabilities equivalent to integrating three to four altimeters, which is consistent with the conclusions drawn by Pujol et al. (2012).



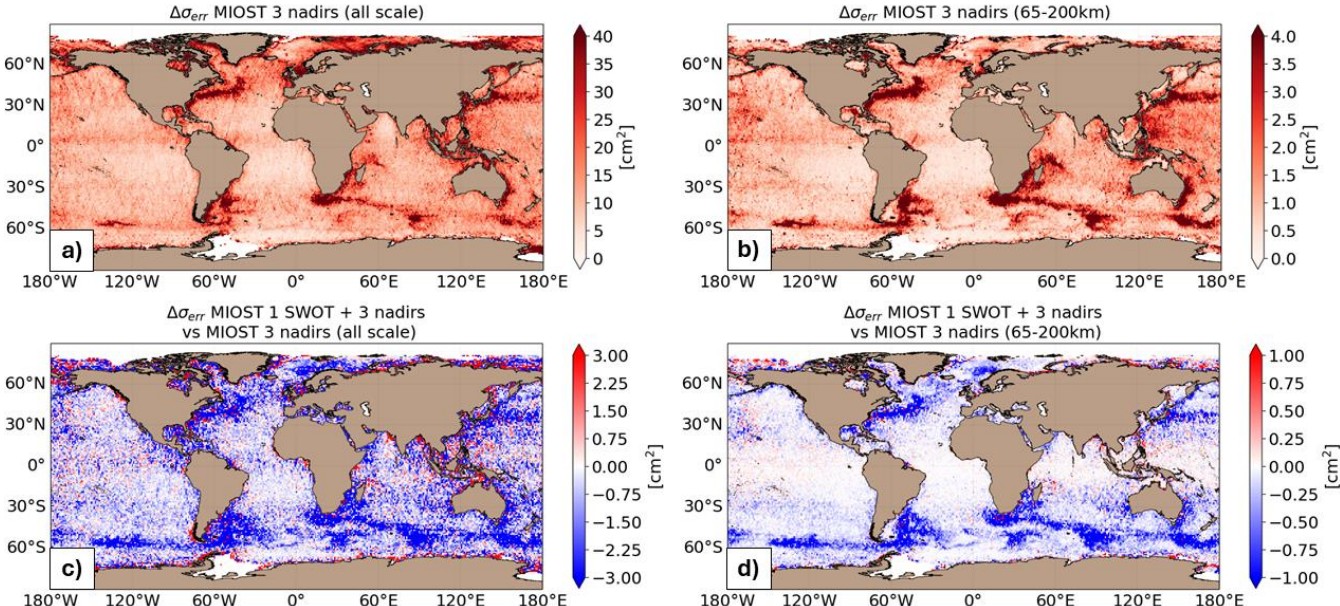

**Figure 6: Variance of the difference SSH$_{map}$-SSH$_{alongtrack}$ computed for the EXP3 (mapping 3 nadirs) and in considering a) all spatial scale and b) spatial scale between 65 km and 200 km. Gain/loss of the mapping error variance of SLA in EXP4 relatively to the EXP3 mapping error variance for c) all spatial scale and d) scale between 65 km and 200 km. Blue colour means a reduction of error variance when SWOT is included in the mapping.**

**Table 4: Regionally averaged mapping error variance for EXP1, EXP3 and EXP4. Score in brackets is the gain/reduction of error variance on the SSH variable between EXP3 and EXP4 and EXP3 and EXP1.**

|  | EXP3: 3 nadirs | | EXP4: 1SWOT + 3 nadirs | | EXP1: 6 nadirs | |
|---|---|---|---|---|---|---|
|  | Err all scale [cm$^2$] | Err 65-200km [cm$^2$] | Err all scale [cm$^2$] | Err 65-200km [cm$^2$] | Err all scale [cm$^2$] | Err 65-200km [cm$^2$] |
| Coastal (< 200km) | 31.8 | 2.5 | 31.2 (-2%) | 2.4 (-4%) | 28.3 (-12%) | 2.2 (-12%) |
| Offshore (> 200km) high variability (> 200cm$^2$) | 32.9 | 5.1 | 25.3 (-23%) | 3.6 (-29%) | 24.2 (-26%) | 3.5 (-31%) |
| Offshore (> 200km) low variability (< 200cm$^2$) | 12.4 | 1.2 | 11.4 (-8%) | 1.0 (-17%) | 11.2 (-10%) | 1.0 (-17%) |
| Equatorial band (10°S-10°N) | 11.7 | 1.1 | 11.3 (-3%) | 1.1 (0%) | 10.6 (-9%) | 1.1 (0%) |


Figure 7a shows the effective resolution of maps generated using three nadir altimeters, while Figure 7b depicts the enhancement in resolution achieved through the integration of SWOT into mapping processes. Maps created with three nadir altimeters have resolutions ranging approximately from 100 km at high latitude to 500 km in the equatorial region. SWOT helps in refining map resolutions at mid-latitudes, locally improving by up to 50 km and on average around 10-20 km enhancement (see Figure 7b). The integration of SWOT in the mapping seems to degrade the resolution in the equatorial band and north Pacific basin.



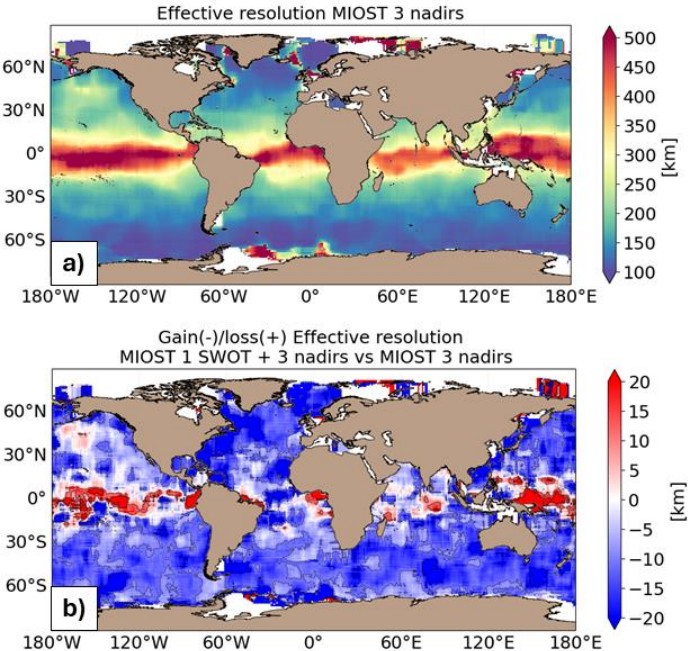

**Figure 7: Maps of effective spatial resolution (in km) for (a) the EXP3 and (b) gain/loss of effective resolution (in km) between EXP4 and EXP3. Blue means finer resolution when SWOT is included in the mapping.**


## 5 Discussion and Conclusions

This study proposes a global analysis of the integration of wide swath altimetry data into an ocean surface topography mapping system. More specifically, our focus has been on quantifying the added value of wide swath data, such as those currently produced by SWOT, within a dense constellation of nadir altimeters (the current constellation comprising 6-7 nadirs) and

within a reduced constellation of 3 nadirs, representative of a future constellation envisioned by 2030 (see Figure 1).

The results obtained in the current nadir altimeter constellation show a moderate contribution of KaRIn data to our MIOST mapping system. Although SWOT helps reduce mapping errors in energetic currents, this reduction remains moderate (10%) because the six nadir altimeters currently in flight already effectively constrain surface dynamics. Consequently, the surface

topography products incorporating SWOT data will benefit from slightly improved spatial resolution (<10 km on average), particularly at mid-latitudes. Some regions where the integration of SWOT appears to alter mapping require better understanding. These regions are characterized by specific atmospheric and oceanic conditions, such as tropical rainfall, wet troposphere, as well as areas affected by storm tracks or internal tides. Improved understanding and processing of KaRIn data in these regions will likely mitigate these deficiencies in future reprocessing efforts.






The European Space Agency (ESA) is currently exploring various combinations of nadir and wide-swath altimeters for the upcoming Sentinel-3 Next Generation (S3-NG) constellation, planned for launch around 2030 (Figure 1). Ongoing studies undertaken in OSSE framework, such as those reported in the papers by Benkiran et al. (2024) and King et al. (2024), aim to provide insights into the impact of these different altimeter constellations on operational ocean forecasts. Based on the experiments conducted for our own study, we propose here to discuss the impact of including a wide-swath altimeter in a constellation of 3 nadirs, which could potentially be operational by 2030, although it does not strictly align with the constellation scenarios envisioned by ESA (either 12 nadirs or 2 wide swaths). The mapping results obtained in a reduced nadir altimeter constellation show more contrasting conclusions about the contribution of KaRIn data in mapping. Indeed, the impact of KaRIn data is more significant in the reduced constellation since only three nadirs constrain the SSH variability less effectively than in a 6-nadir constellation. With SWOT, the mapping errors are reduced by ~30% in energetic regions and the gain in effective resolution at mid latitude reaches more than 20km. Additionally, the comparison between the 1SWOT+3 nadirs and 6 nadirs experiments suggests that integrating a SWOT mission into Level 4 products could offer an observational capacity equivalent to integrating three to four altimeters, in line with the conclusions drawn in an OSSE framework by Pujol et al. (2012) and Bellemin-Laponnaz et al. (2022). It is also important to emphasize that we currently have only one SWOT instrument in the altimetric constellation, resulting in a significant imbalance between high spatial resolution and low temporal resolution in certain regions. This gap is largely addressed within the framework of S3-NG when considering scenarios involving two wide-swath instruments. Furthermore, it is relevant to stress the preliminary nature of this initial mapping attempt, and additional studies within the scenarios envisaged in the S3NG project will need to be conducted and rigorously analyzed with OSSEs. The first experiments aiming to quantify the contribution of wide-swath altimeter data were conducted within the framework of an OSSE. These OSSEs have allowed both to prepare the mapping system for Karin data and to assess the contribution of Karin data to mapping. Even though OSSEs may be based on simplifying assumptions, we believe they provide a valuable framework for quantifying mapping performance under different observing system scenarios. We present a comparison of the contribution of Karin data in the OSE and OSSE frameworks in the Appendix A, demonstrating satisfactory agreement between the results of each experiment and the crucial role of OSSEs in guiding decisions concerning the design of future altimeter constellation configurations.

Furthermore, the global experiments presented here were carried out with the MIOST mapping system. It is worth noting that the MIOST mapping approach represents one of several potential methods for integrating SWOT data into Level 4 gridded products and improve the space/time resolutions of the maps. Our study shows that refinement in mapping method is required to effectively map fine scale ocean structures. Various alternative approaches have been explored to retrieve finer oceanic structures, encompassing assimilation of altimetric data into both simple ocean models (e.g., BFN-QG, Le Guillou et al., 2021a, 2023; 4DVAR-QG, Le Guillou et al., 2024) or dynamic interpolation method (Ubelmann et al, 2015, Ballarotta et al., 2020), and more complex ocean models (Benkiran et al., 2021, Tchonang et al., 2021; Archer et al., 2022, Benkiran et al., 2024; King et al., 2024, Souopgui et al., 2020; Zhou et al., 2024). Additionally, data-driven techniques (Fablet et al., 2021, Beauchamp et



al., 2023, Martin et al., 2023, 2024, Archambault et al., 2024), and the Local polynomial fitting approach by Lilly (2023), have been examined for spatial and temporal interpolation of altimetric data. Certain of these methods have demonstrated advantages in effectively mapping the complex structures of turbulent and intense ocean currents at fine scales and proved to be very efficient for accounting the imbalance between the high spatial resolution of SWOT data and its sparse temporal sampling at certain latitude. Preliminary experiments aimed at exploring the potential of SWOT in 4Dvarnet and 4DvarQG

systems have just been conducted and appear to indicate that these systems are capable of better representing the nonlinear dynamics of ocean surface topography. Appendix B focuses on the capability of the 4DvarQG and 4DvarNET mapping approaches to capture finer structures. These alternative mapping methods have been tested at the regional scale, and initial analyses of their results show relatively good performance compared to the MIOST reconstruction, with significant gains in effective resolution of more than 20 km compared to equivalent MIOST products. These studies show that using dynamical

constrains in the mapping procedure help to improve the space/time resolutions of the maps. Overall, the 4DvarNET and 4DvarQG methods seem to be good alternative mapping approaches of the MIOST solution. The current constraint with both the 4DvarNET and 4DvarQG methodologies is their inability to offer a global-scale solution for mapping SSH but efforts are underway to address this limitation.

Finally, it is important to mention that the ocean surface circulation results from a complex interplay of phenomena occurring across various spatial and temporal scales, ranging from the slowly evolving large meso-scale eddies to more rapidly changing surface waves, submesoscale eddies, filaments, and fronts (see e.g. Figure 3 in Chelton, 2001). Mesoscale eddies have been studied for over 30 years using multi-mission gridded altimetry products and are recognized as key drivers in the horizontal transport of heat, nutrients, and carbon (Wolfe et al., 2008; Klein et al., 2009; Griffies et al., 2015, Dong et al, 2014). In

contrast, submesoscale variability and its role in ocean dynamics remains poorly observed and understood. These smaller-scale eddies may also play a crucial role in turbulent transport, mixing, and energy dissipation, and their impact on ocean dynamics may be particularly significant in the context of climate change, as they influence the ocean's uptake of heat and $CO_2$ (Zhang et al., 2023). Our analysis indicates that integrating high-resolution ocean surface observations with dynamically constrained mapping approaches could significantly enhance the accuracy of operational sea-level gridded products. From a

physical oceanography perspective, these refined products are likely to improve our analysis of coherent vortices, increase our understanding of fine-scale ocean processes, which is essential for improving climate models and predictions. An analysis of the relative vorticity fields (detailed in Appendix B) reveals, for example, significant differences between mapping methods, particularly in the small-scale structures, fronts, and filaments which seems to be more pronounced in the 4DvarQG solution than the MIOST and 4DvarQG solutions. From an operational oceanography perspective, the refined products would

participate in enhancing decision-making processes related to ocean safety, marine pollution management, ship routing, and the sustainable utilization of fishing resources. By providing more accurate data, these products would enable more effective responses to environmental challenges and contribute to the overall safety and sustainability of maritime activities.





### A - Discussion on the robustness of Observing System Simulation Experiments

Before the launch of SWOT and the acquisition of the first KaRIn data, only Observing System Simulation Experiments (OSSEs) have been conducted to study the contribution of wide swath in mapping systems (Benkiran et al., 2021, Tchonang et al., 2021, Ubelmann et al., 2015, Ballarotta et al., 2020, Le Guillou et al., 2021, Bellemin Laponnaz et al., 2022). An OSSE is a method to assess the potential impact of new observing systems or changes to existing ones on data assimilation or mapping systems. In an OSSE, synthetic observations are generated to simulate the data that would be collected by a proposed observing

system. These synthetic observations are then integrated into mapping systems to evaluate the impact on the resulting analyses or forecasts. By comparing the results from the OSSE with and without the proposed observing system, one can assess the potential benefits and limitations of the new observations. Yet, questions arise regarding the reliability and validity of studies conducted within the idealized framework of OSSEs. We propose here to discuss about the robustness of the results obtained in OSSE with SWOT KaRin data.


To prepare the MIOST mapping system for real KaRIn data, we conducted several OSSEs based on a one-year-long realistic ocean numerical simulation. Specifically, we used SSH data from the global simulation GLORYS12v1 (Lellouche et al., 2021). The GLORYS12v1 reanalysis is a global ocean dataset generated by assimilating historical observations into the NEMO ocean model, driven at the surface by ECMWF ERA-Interim data. This reanalysis covers the entire globe and is provided on a 1/12°

regular grid.

We produced three datasets for nadir altimeters by interpolating model outputs onto each mission ground track. Similarly, we employed the SWOT Simulator (Gaultier et al., 2016) for generating SWOT data. Subsequently, we conducted two mapping experiments: one using the 3-nadir constellation and one using the 3-nadir constellation + 1 SWOT as input data, similar to the EXP3 and EXP4 experiments previously described in the paper within an OSE framework.

These OSSEs are based on several simplifying assumptions, one can mention:

    1) Measurement noise applied to the data is idealized (for example, a 3cm standard deviation Gaussian noise is added to the simulated nadir data).

    2) Some long-wavelength biases present in real data have not been accounted for in the simulated data.

    3) While real data is sometimes edited due to poor quality, no editing is applied to the simulated data.

4) In this GLORYS12v1 simulation, barotropic tides and internal tides are not explicitly resolved. Therefore, the pseudo-observations extracted from this simulation do not capture the residuals of these signals, unlike real altimetric data.

Despite these assumptions, the comparison of mapping enhancements resulting from the integration of the SWOT Karin instrument demonstrates relatively good agreement between the OSSE and OSE frameworks, as depicted in Figure A1. The

areas where the integration of SWOT into OSEs and OSSEs leads to error reductions are generally similar, although the extent of these improvements may vary locally. Specifically, notable differences between OSSEs and OSEs appear to manifest in



certain regions, where it is likely that the content of simulated data is too idealized within the chosen OSSEs (for example due to the absence of internal wave residuals). Therefore, the use of more realistic numerical simulations, such as those integrating tidal waves in very high-resolution simulations like eNATL60 (Brodeau et al., 2020) or MITgcm (Marshall et al., 1997; Su et

al., 2018), seems to be a better option for designing OSSEs. Overall, largest error reductions when incorporation KaRin are found in regions of high SSH variability (western boundary currents, ACC). Same conclusions are drawn concerning the effective resolution derived from OSE and OSSE framework (Figure A2). These good agreement between OSE and OSSE results seems to indicate that OSSEs can plays a significant role in shaping decisions regarding future altimeter constellation scenarios.

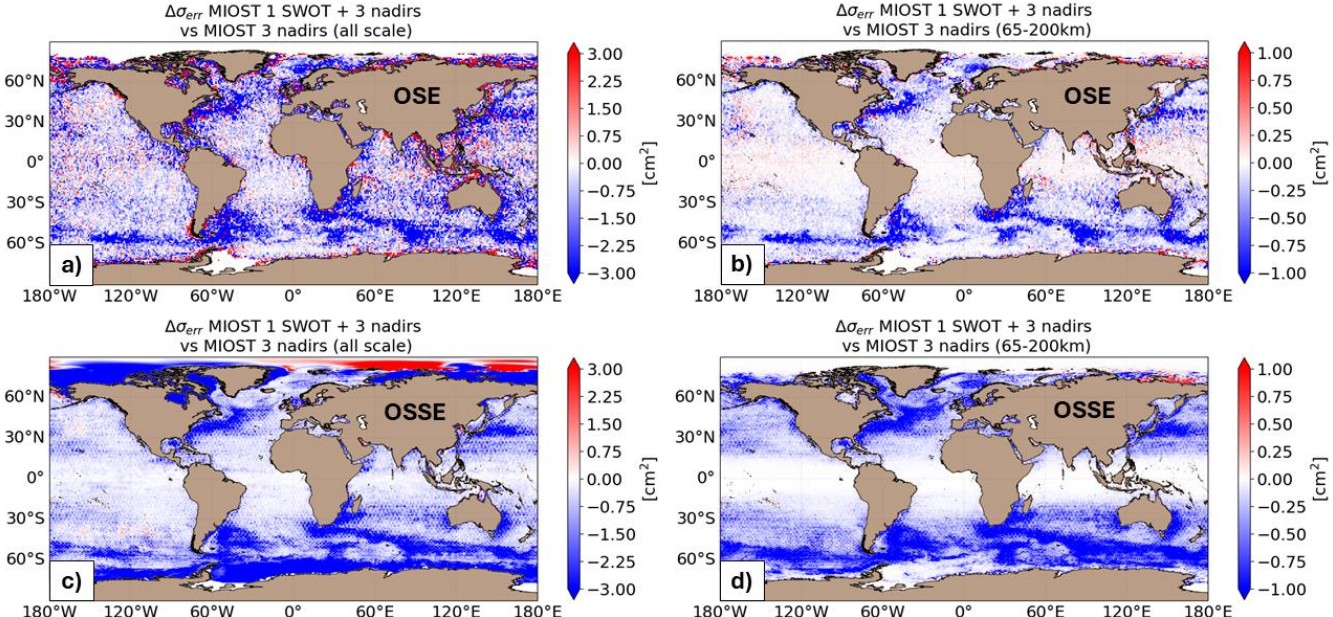


**Figure A1: Gain/loss of the mapping error variance of SLA when integrating SWOT into a 3 altimeters constellation for a) all spatial scale in OSE, b) scale < 200km in OSE, c) all spatial scale in OSSE, and d) scale < 200km in OSSE. Blue colour means a reduction of error variance when SWOT is included in the mapping.**



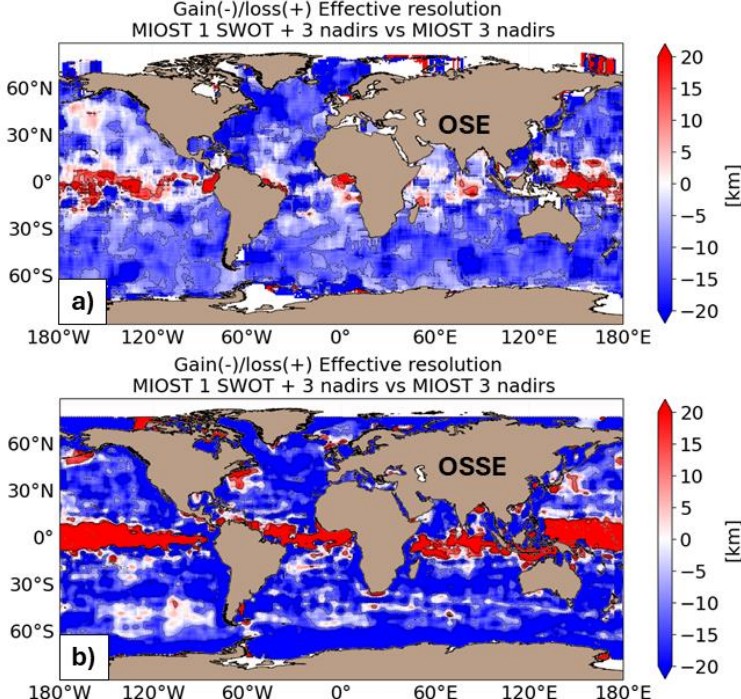

**Figure A2: Maps of gain/loss of effective resolution (in km) in integration SWOT in a 3 altimeters constellation in a) OSE and b) in OSSE. Blue means finer resolution when SWOT is included in the mapping.**

## B – Alternative mapping methods

Various alternative mapping approaches exist to retrieve finer oceanic structures. We propose here to illustrate the performance of two alternative mapping methods that we specifically tested for our study: one based on an assimilated quasi-geostrophic model (4DvarQG), and another based on a data-driven approach (4DvarNET)

The 4DvarQG mapping technique[1] (Le Guillou et al., 2024) integrates a 4-Dimensional variational (4DVAR) scheme with a Quasi-Geostrophic (QG) model. The method integrates a weakly constrained, reduced-order, four-dimensional variational (4DVAR) scheme with a 1.5-layer quasi-geostrophic (QG) model. An error term is optimized to align the QG dynamics with the observations. This term is projected onto a reduced basis consisting of space-time wavelets to ensure the convergence of the optimization process. The structure of the wavelet elements and their expected variances are carefully selected to represent the SSH variability of the AVISO/DUACS multi-mission product.

---

[1] https://github.com/leguillf/MASSH





The 4DvarNET mapping algorithm[2] (Fablet et al., 2021, Beauchamp et al., 2022) is a data-driven approach combining a data assimilation scheme associated with a deep learning framework. This neural network framework involves the joint training of the representation of the ocean dynamic, as well as of the solver of the data assimilation problem. The 4DvarNET algorithm is trained using a supervised learning strategy in an OSSE context, taking the SSH variable of an ocean model as ground truth. Once trained in OSSE, the 4DvarNET algorithm is ready to perform SSH reconstructions with real altimetric data as input.

For our study, 4DvarNET was trained on the eNATL60-BLB002 realistic high-resolution simulation[3] over a portion of the Gulfstream region ([32°N-47°N] [66°W:51°W]). Pseudo-observations were generated from this numerical simulation to represent the present-day nadir altimeter constellation as well as KaRin swath.

Specific experiments similar to EXP1 and EXP2 (cf. Table 2) were carried out over the North Atlantic basin ([25°N:50°N]
[80°W: 10°W]) to estimate the impact of SWOT swath and the performance of each method relative to the MIOST mapping approach.

Figure B1 & B2 focuses on the impact of KaRin data in each method. The KaRIn data primarily enables the reduction of mapping errors within the main Gulf Stream current. The integration of KaRIn into these mapping algorithms results in an average error reduction of 0.5 cm² for the 4DVARQG method, 0.5 cm² for the 4DVARNET method, and 0.4 cm² for the
MIOST method. In terms of effective resolution gain, the impact of KaRIn varies depending on the mapping methods employed: the gain is moderate (<10 km) with the 4DVARQG method, more significant (10-20 km) with 4DVARNET, and intermediate with MIOST (~10 km). This suggests that linear interpolation methods such as MIOST or model dynamics-based methods like 4DvarQG are less sensitive to dense KaRin observations compared to the data-driven method 4DvarNET, which manages to exploit more content from the KaRin swath.

---

[2] https://github.com/CIA-Oceanix/4dvarnet-core

[3] https://github.com/ocean-next/eNATL60



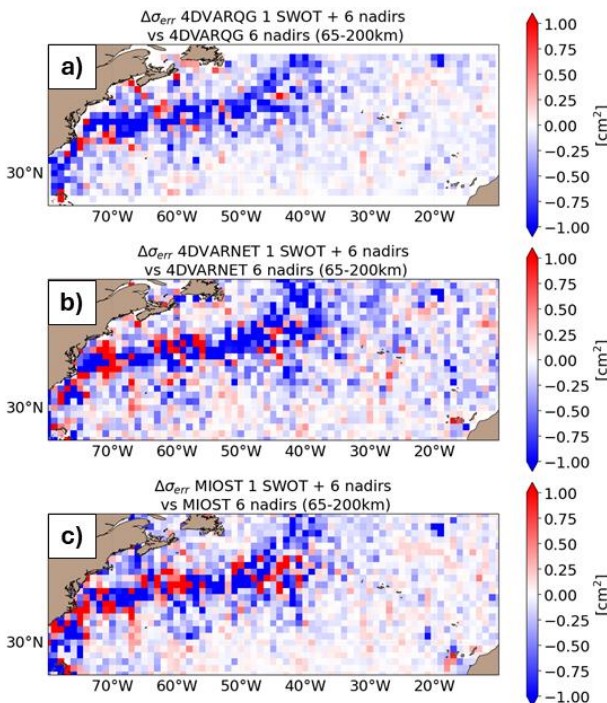


**Figure B1: Gain/loss of the mapping error variance of SLA in EXP2 relatively to the EXP1 mapping error variance for scale between 65 km and 200 km in a) the 4DVARQG mapping method, b) the 4DVARNET mapping approach and c) the MIOST approach. Blue colour means a reduction of error variance when SWOT is included in the mapping.**



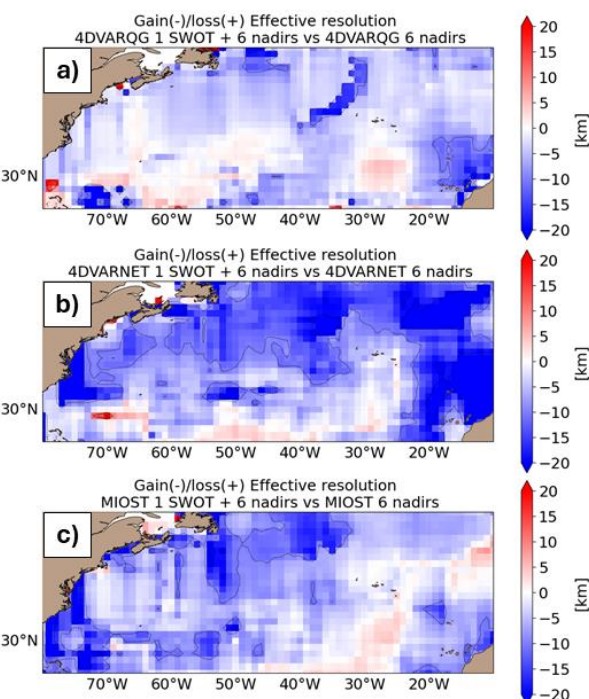

**Figure B2: Gain/loss of effective resolution (in km) between EXP2 and EXP1 for a) the 4DVARQG mapping method, b) the 4DVARNET mapping approach and c) the MIOST approach. Blue means finer resolution when SWOT is included in the mapping.**

Figure B3 and B4 intercompare the performance of the 4DVARnet and 4DvarQG methods relatively to MIOST approach. These intercomparisons demonstrate that the 4DvarQG and 4DvarNET methods provide better surface topography reconstructions than the MIOST method, particularly in the main pathway of the Gulf Stream where nonlinear energetic dynamics dominate (see Figure B3a and b). The average error reduction is 1.9 cm² for the 4DVARQG method and 0.5 cm² for the 4DVARNET method compared to the MIOST solution. In regions of low oceanic variability, the 4DvarQG and MIOST methods are relatively equivalent, whereas the 4DvarNET method appears to degrade the solution in these areas, likely due to the fact that the 4DvarNET model was trained only on a limited region of the Gulf Stream. This results in a significant improvement in resolution with the 4DvarNET and 4DvarQG methods, particularly in the main current of the Gulf Stream, where resolution gains of over 20 km are observed. The comparison between the 4DvarNET and 4DvarQG methods shows that the dynamical 4DvarQG method is able of mapping finer scales than the 4DvarNET in this part of the North Atlantic basin.





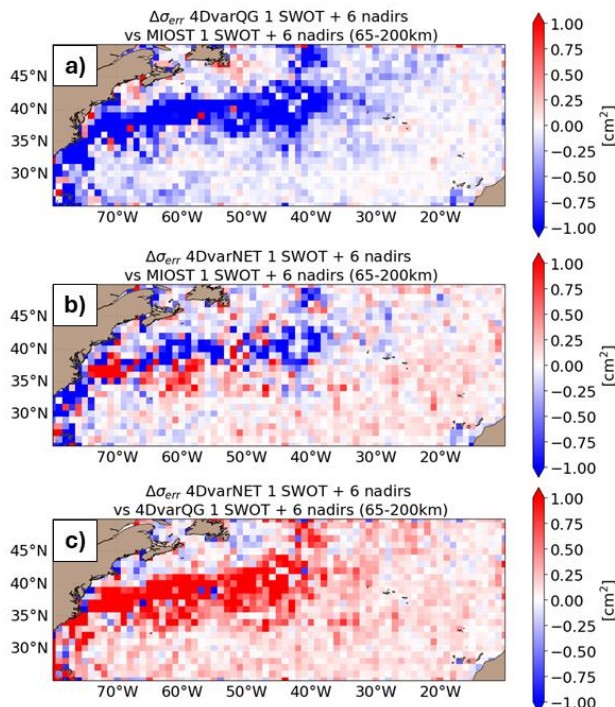

**Figure B3: Gain/loss of the mapping error variance (for spatial scale < 200km) of SLA in EXP2 between 4DvarQG and MIOST (blue colour means error reduction with 4DvarQG) b) between 4DvarNET and MIOST (blue colour means error reduction with 4DvarNET) and c) between 4DvarNET and 4DvarQG (blue colour means error reduction with 4DvarQG)**





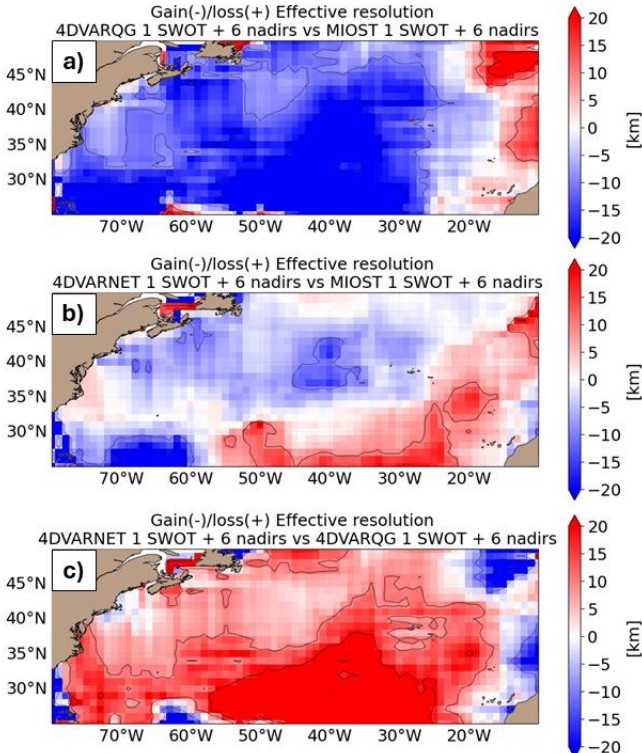

**Figure B4: Gain/loss of effective resolution (in km) between a) the 4DVARQG and MIOST mapping methods (Blue means finer resolution with 4DvarQG), b) the 4DVARNET and MIOST mapping methods (Blue means finer resolution with 4DvarNET) and c) the 4DvarNET and 4DvarQG mapping method (Blue means finer resolution with 4DvarNET)**

Qualitative assessments can also highlight differences between each method, particularly when analyzing the relative vorticity maps derived from the SSH gridded field. Figure B5 presents the relative vorticity maps on August 30, 2023 over the Gulf Stream region for each method. There are notable disparities in the structure of the relative vorticity fields, especially in the small-scale structures, fronts, and filaments, which are more pronounced in the 4DvarQG solutions. These features, crucial for surface energy transfer, horizontal and vertical transport, and heat and carbon uptake (Wolfe et al., 2008; Klein et al., 2009; Griffies et al., 2015), are difficult to map accurately with current operational mapping algorithms and appear more realistic and coherent with the 4DvarQG approach.



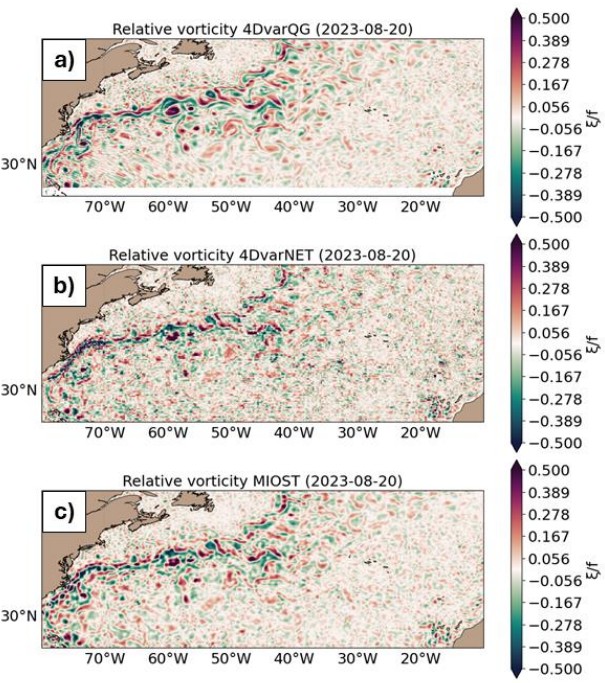


**Figure B5: Relative vorticity snapshot for August 20th, 2023 with a) the 4DvarQG reconstruction, b) the 4DvarNET reconstruction and c) the MIOST reconstruction**

Overall, the 4DvarNET and 4DvarQG methods seem to be good alternative mapping approaches of the MIOST solution. The current constraint with both the 4DvarNET and 4DvarQG methodologies is their inability to offer a global-scale solution for mapping SSH but efforts are underway to address this limitation.


**Author contributions:**

MB designed the experiments, conducted the MIOST mapping and analysis, and wrote the manuscript with contributions from co-authors. VBL carried out the initial work on OSSEs. GM, CA, and AL handled the 4DvarNET mapping and analysis, while FLG conducted the 4DvarQG mapping and analysis. AD optimized the MIOST code. GD, YF, RF, and MIP got funding and managed the project. All authors reviewed and edited the draft version.


**Competing interests:** The authors declare no competing interests.




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

**Data availability**



The SWOT_L3_LR_SSH product, derived from the L2 SWOT KaRIn low rate ocean data products (NASA/JPL and CNES), is produced and made freely available by AVISO and DUACS teams as part of the DESMOS Science Team project". AVISO/DUACS, 2024. SWOT Level-3 KaRIn Low Rate SSH Expert (v1.0) [Data set]. CNES.

https://doi.org/10.24400/527896/A01-2023.018

The Near-Real-Time (NRT) Level-3 altimeter satellite along-track data are distributed by the EU Copernicus Marine Service (product reference SEALEVEL_GLO_PHY_L3_NRT_008_044, Pujol et al., 2023).

The Gridded Sea Level Height and geostrophic velocities products computed with nadirs & wide swath altimetry and presented in this study are made freely available by AVISO and DUACS teams as part of the DESMOS Science Team project. These products were processed by SSALTO/DUACS and distributed by AVISO (https://www.aviso.altimetry.fr) supported by CNES. DOI: 10.24400/527896/a01-2004.007

Specific maps (excluding Saral/Altika) made for this study are made available in a collaborative data-challenge: https://github.com/ocean-data-challenges/2024_DC_SSH_mapping_SWOT_OSE