# Peer review of "Integrating wide swath altimetry data into Level-4 multi-mission maps"

_EGUsphere, 2024_

## Author Response (AR1)

Referee #1 comments

This work focuses on integrating high-spatial-resolution sea surface height (SSH) measurements from the SWOT mission into high-level gridded SSH maps derived from a constellation of nadir altimeters. The integration method is based on the Multiscale Inversion of Ocean Surface Topography (MIOST) technique (Ubelmann et al., 2021, 2022). In addition to data integration, the paper explores hypothetical scenarios for future wide-swath mission development and assesses the trade-offs between nadir altimeters and SWOT-type wide-swath altimeters. The study concludes that while SWOT's contribution is moderate with six nadir altimeters, it becomes substantial if the number of nadir altimeters is reduced to three, suggesting potential future configurations. The study also highlights the promise of several alternative data-driven methods for gridding.

The research is well-designed and thoroughly executed, with clear and well-structured writing. Here I have a major comment which only involves adding some discussion in the abstract and a minor suggestion in using SWOT data.

Major comment:

SWOT has demonstrated exceptional performance, with extremely low noise levels, capturing SSH signals beyond the geostrophic flow. However, this study focused on signals larger than 65 km, meaning the full potential of SWOT was not fully realized. This limitation is understandable, as neither the methodology nor the validation data is designed to capture signals below 65 km. As a result, the conclusion regarding SWOT's contribution is only partially realized, as the existing gridding framework and validation data impose constraints. It could be a missed opportunity if we focus solely on gridding geostrophic information, which represents only a small portion of SWOT's advancements. I recommend the authors include this discussion in the abstract and highlight that new innovations are needed to fully harness SWOT's potential beyond geostrophic surface flow, which include nonlinear eddy dynamics as well as linear and nonlinear internal waves. Those small-scale phenomena are non-retrievable in the nadir altimeter regardless of the number of conventional satellites. This is important factor to consider in discussing advantages and disadvantages of including SWOT in the future constellation. I do not believe that we have yet setup the data gridding or assimilation framework to address this question in an OSE environment.

Minor suggestion:

Since the validation relies on withheld nadir altimeter data and lacks reliable small-scale (<65 km) ground truth, I suggest applying a low-pass filter to SWOT data before integrating it into the MIOST for gridding. This would effectively remove high-amplitude internal tides, solitary waves, and small-scale residual mean sea surface variations from SWOT, thereby improving the accuracy of the MIOST gridding, especially in low latitudes where internal solitons are prevalent. In these regions, the inclusion of unfiltered SWOT data tends to degrade the mapping accuracy as shown in the manuscript.

Answer to Referee #1

Dear Jinbo Wang,

Thank you for your feedback on our manuscript. We greatly appreciate the time you have taken to review our work. In response to your comments, we have made several updates to the manuscript, particularly in the Discussion & Conclusion sections, to better address the points you raised.

*Major comment*: We fully agree with your comment regarding the potential of SWOT in capturing smaller-scale ocean dynamics beyond geostrophic flow. To address this, we have revised the Discussion & Conclusion section to explicitly acknowledge that the current study focuses on signals larger than 65 km and, as a result, does not fully leverage SWOT's ability to capture smaller-scale phenomena such as nonlinear eddy dynamics and internal waves. We now discuss the importance of developing new

frameworks that can harness SWOT's full potential, especially for capturing submesoscale dynamics that cannot be retrieved by nadir altimeters.

We have also modified the abstract to reflect this limitation more clearly and to highlight that while our current methodology emphasizes larger-scale geostrophic flows, future research must consider the need for innovations in data gridding and assimilation to fully exploit SWOT's capabilities.

*Minor comment:* Thank you for your suggestion about applying a low-pass filter to the SWOT data prior to its integration into the MIOST for gridding. We will evaluate the possibility of using a low-pass filter or directly incorporating the filtered Karin SSHA product to improve the accuracy of the MIOST gridding process. We appreciate your recommendation and will consider it in our future reprocessing efforts as we work to refine our methodology.

Thank you again for your valuable feedback, and we look forward to any further suggestions you may have.

Review of "Integrating wide swath altimetry data into Level-4 multi-mission Maps", by Ballarotta et al.

This paper describes an application that combines SWOT data and nidar altimeter data. This is one of the first to combine these data and demonstrates very good results. This is very encouraging. The figures in this paper are excellent, and the analysis is thorough and clear. The authors separate analysis by scale, making it more meaningful to more readers. I would suggest that this is acceptable in the current form, but I think it would benefit from a bit more detail in the methods section. The authors refer readers to a different paper, where the method is described and assessed. I think it'd be better to include sufficient details in this paper that a reader can read this paper in isolation.

Detailed comments

L62: "only through OSSEs". This is not quite true. An exception is Lui et al. (2024; Frontiers in Marine Science; doi: 10.3389/fmars.2024.1456205).

**Authors Response ()AR: You are right, an increasing number of studies are currently underway to access the impact of real SWOT data on mapping systems. The paper you are mentioning is indeed also one of the pioneering studies looking at this impact of real swot data.**

Figure 2: is excellent

**AR: Thank you !**

Section 2.2 is too brief. A reader of this paper cannot understand the method (MIOST) from the description offered. Consider providing a more complete description and quoting some details of the expected error. I took some time to look at the Ballarotta et al. (2023) paper, because I did not know about MIOST. I can see that the system performance is good – better than the well-established DUACS system. I think it'd be better to include enough information in this paper so a reader doesn't have to go to another paper to find the relevant information.

**AR: Section 2.2 have been expanded to provide more description of the MIOST mapping system.**

L155: I like the separation of scales for assessment.

**AR: Thank you !**

Eq (2) and (3): I wonder if sigma squared would be better, since you report variances. But more generally, I wish you'd use error standard deviation so the units are more easily understood (in m, instead of m^2, for example).

**AR: All the metrics have been updated using the error standard deviation with unit in cm.**

Table 2 defines the experiments (EXP1, EXP2 etc), but these names are not used in Figure 3. I prefer the names used in Figure 3. Maybe you needn't give the experiments meaningless names (EXP1, EXP2 etc).

**AR: EXP1, 2 etc are now mentioned by their names used in the Figures**

Figure 3, 4, 6: "X vs Y" is unclear. I presume this is the difference between two experiments (X – Y). Can you make this clear?

**AR: This has been corrected in the updated version of the manuscript. Now the difference is X – Y**

Figure 3: is beautiful

**AR: Thank you !**

L283: I would regard an improvement of 10% very good – not modest. Recall that we have taken decades to learn how to fully exploit the nadir altimeter data. This demonstrates that we can immediately improve this by 10% at (one of ) the first serious attempt. I like that the authors are stating this conservatively. But perhaps they are under-stating the improvement.

Dear Peter Oke,

We sincerely thank you for your positive feedback and constructive suggestions on our manuscript. We are pleased to hear that you liked our manuscript.

In response to your suggestions, we have made several updates to enhance the paper's clarity. Specifically, we have expanded Section 2.2 to provide a more detailed description of the MIOST method, including key details on error expectations, so that readers can better understand our approach without needing to refer to previous studies.

We also revised the Figures to explicitly display the error as standard deviation (rather than variance) for improved interpretability. Titles and labels have been adjusted to clarify that these figures represent differences between experiments. Additionally, we have incorporated clearer labels for the experiments (rather than using "EXP1" or "EXP2"), as you recommended, to facilitate better understanding for readers.

Thank you again for your thoughtful review and for highlighting areas where further clarification could benefit the readership. We believe these revisions address your concerns and enhance the overall quality of our work.

Best regards,

---

## Author Response (AR2)

Dear Ilker Fer,

Thank you for your message for accepting our manuscript for publication in the journal Ocean Science. We have updated the manuscript following the minor corrections you mentioned.

Sincerely,

Maxime Ballarotta